# Effectiveness of Interventions for Reducing Sedentary Behaviour in Older Adults Living in Long-Term Care Facilities: A Protocol for a Systematic Review

**DOI:** 10.3390/healthcare11141976

**Published:** 2023-07-08

**Authors:** Erika Karkauskiene, Mark A. Tully, Vilma Dudoniene, Maria Giné-Garriga, Anna Escribà-Salvans, Cristina Font-Jutglà, Javier Jerez-Roig

**Affiliations:** 1Department of Health Promotion and Rehabilitation, Lithuanian Sports University, 44221 Kaunas, Lithuania; vilma.dudoniene@lsu.lt (V.D.); javier.jerez@uvic.cat (J.J.-R.); 2School of Medicine, Faculty of Life and Health Sciences, Ulster University, Londonderry BT48 7JL, UK; m.tully@ulster.ac.uk; 3Blanquerna Faculty of Psychology, Education and Sport Sciences, Ramon Llull University, 08022 Barcelona, Spain; mariagg@blanquerna.url.edu; 4Blanquerna Faculty of Health Sciences, Ramon Llull University, 08022 Barcelona, Spain; 5Department of Social Sciences and Community Health, Research Group on Methodology, Methods, Models and Outcomes of Health and Social Sciences (M_3_O), Faculty of Health Sciences and Welfare, Centre for Health and Social Care Research (CESS), University of Vic-Central University of Catalonia, 08500 Vic, Spain; anna.escriba1@uvic.cat (A.E.-S.); cristina.font1@uvic.cat (C.F.-J.)

**Keywords:** aged, sitting position, risk factors, cohort studies

## Abstract

*Background.* Sedentary behaviour (SB) is an important risk factor for several health-related outcomes. The prevalence of SB is alarmingly high in older adults, who spend on average 9.4 h being sedentary each day, making them the most sedentary of all age groups. *Objectives.* The primary objective of this review is to assess the impact of interventions aimed at reducing SB in older adults (aged 60 years and older) living in long-term care facilities (LTCFs). The research question for this systematic review is as follows: in older people living in LTCFs, do interventions aimed at reducing SB, compared to usual care, result in a decrease in SB daily time or a reduction in the length of prolonged and uninterrupted sitting bouts? *Data sources.* Only peer-reviewed articles will be included in this systematic review, articles will be identified using the PICO method in seven different databases. *Participants and interventions.* Any primary intervention study (including randomized controlled trials, non-randomized controlled trials, and cohort studies) with the aim to reduce SB daily time or shorten the length of prolonged and uninterrupted sitting bouts in older adults living in LTCFs will be included. After searching databases, abstracts of the studies will be screened, and, after retrieving full text articles, data extraction will be conducted by two independent reviewers. *Study appraisal and synthesis methods.* The review will adhere to PRISMA reporting guidelines. Risk of bias (RoB) will be assessed using ROBINS-I or the RoB 2.0 tool and will be discussed with a third reviewer. The data will be grouped according to study design, with separate analysis for randomised and non-randomised designs. *Results*. The primary outcomes will be SB or time spent sedentary, assessed before and after the intervention. For the outcomes with the same measurement units, the pooled mean differences will be calculated. Standardised mean differences will be calculated for the outcomes with different measurement units. The data not suitable in numbers will be synthesised narratively. The strength of evidence of the outcomes will be assessed using GRADE assessment. If the data are suitable for quantitative analysis, we plan to use the Revman software to conduct a meta-analysis. *Conclusions and implications of key findings.* This protocol can serve as a valuable resource for other researchers interested in conducting similar systematic reviews or meta-analyses in the field of SB and older adult health.

## 1. Introduction

Inactivity and sedentary behaviour (SB) have been recognized as major health risk factors, particularly among older adults. SB is defined as any waking activity using less than 1.5 metabolic equivalents (METs) while in a seated, reclining, or lying position [1]. Studies have consistently linked unfavourable body composition and cardiovascular risks to physical inactivity, particularly to taking less than 5000 steps per day. As a result, researchers and practitioners consider it appropriate to use the 5000-steps-per-day threshold as a measure of being sufficiently active and 8-or-more-hours-per-day threshold as a measure of having sedentary behaviour [2].

### 1.1. SB in the Population of Older Adults

The prevalence of SB is alarmingly high in older adults, who spend on average 9.4 h being sedentary each day, making them the most sedentary of all age groups [3,4]. According to research, older adults commonly engage in sedentary activities, such as watching television, reading, eating meals, using the computer or transportation [5]. Another study conducted using time-lapse cameras to monitor SB among older adults found that they tend to sit for prolonged periods in the afternoon and evening compared to the morning, and they do so most frequently when they are alone at home. In fact, the study also revealed that older adults spent 70.1% of their sedentary time at home, with 56.9% of this time spent being alone. Additionally, 46.8% of this time was found to occur in the afternoon [6]. According to a recent review, the impact of SB on health can differ based on the specific type of behaviour, and watching TV has been found to have the most negative health-related consequences; this could be because watching TV is a form of passive SB and often involves snacking while doing so [7]. These findings provide valuable insights into the patterns of SB among older adults.

A systematic review conducted by Compernolle et al. in 2020, which examined older adults’ perceptions of SB, identified several reasons why they tend to spend a lot of time sitting [8]. According to this review, physical limitations, pain, loss of mobility, and illness or chronic health conditions were cited as significant factors that contribute to SB. Additionally, mental health problems such as depression, anxiety, and low self-esteem were found to be common barriers to reducing sedentary time among older adults. The review also revealed that many older adults lack knowledge about the benefits of reducing SB and do not receive enough encouragement to break up long periods of sitting [8]. In the systematic literature review conducted in 2015 by Chastin et al., the authors identified various factors that influence SB in older adults, such as age, gender, education level, physical health, psychological factors, and environmental and social factors [9]. Addressing these factors may require a multidisciplinary approach, including targeted education and communication efforts to promote awareness of the importance of reducing SB, as well as interventions that correspond to the specific needs and constraints of older adults.

Studies have established a clear link between prolonged sitting, uninterrupted sitting bouts, and physical inactivity and various negative health outcomes. A recent study conducted in 2020 by Park et al. shows that SB can cause a higher risk of mortality from all causes, cardiovascular diseases, and cancer. Moreover, it can also increase the likelihood of developing metabolic disorders such as diabetes mellitus, hypertension, and dyslipidemia, as well as musculoskeletal issues such as arthralgia and osteoporosis [7]. In addition, the findings of a cohort study by Gilchrist et al., conducted in 2020, involving 8002 middle-aged and older US adults further suggest that increased sedentary time, as measured by accelerometery, is independently associated with a higher risk of cancer mortality. The study also found that replacing sedentary time with light or moderate-to-vigorous physical activity (PA) could lower cancer mortality risk [10].

Furthermore, in relation to SB, two studies report that higher levels of SB are associated with higher levels of sarcopenia [11,12]. Physical inactivity contributes to the development of sarcopenia, either due to disease-related immobility or disability, or to a sedentary lifestyle, which has been shown to be a risk factor for muscle weakness [13]. A 2004 study by Jones et al. demonstrated that the decline in muscle strength and muscle overuse characteristic of sarcopenia in aging, in addition to age, was primarily caused by physical inactivity: within two weeks of limb immobilization, participants had lost 4.7% of quadriceps lean mass and 27% of isometric strength [14].

Insufficient PA and SB are behavioural factors that are directly associated with sarcopenia and sarcopenic obesity [15,16,17]. These behavioural factors can be modified [16]. It is reported that physical inactivity in the older people acts as a catalyst of bone and body composition alterations accelerating the aging process [18]. Associated in SB, large people with a limited walking speed present a higher risk of sarcopenia [16]. Another study revealed that every additional hour of SB per day was associated with 1.06 times higher odds of developing sarcopenia, which is characterized by low skeletal muscle mass and either a slow gait speed or a weak handgrip strength [11].

In addition to affecting physical health, SB influences psychosocial wellbeing and can lead to depression and cognitive impairment [8]. The results from the SITLESS study, a European-wide study of community-dwelling older adults, suggest that higher levels of SB are associated with an increase in the level of social isolation [19]. The results from Shuyun et al., 2021, suggest that elderly people who have greater subjective and social well-being tend to sit less and participate in more physical activities. This tendency was most noticeable among the oldest individuals (aged 80 years and older) [20].

### 1.2. SB among Older Adults Living in Long-Term Care Facilities

Long-term care facilities (LTCFs) are defined as facilities that provide accommodation, support, and assistance with activities of daily living together with intensive forms of care for older and frail people [21]. It has been noticed that the SB among residents living in LTCFs is even more concerning, as they spend an average of 79% of their waking daily time sedentary, mostly partaking in activities such as sleeping, doing nothing, and watching TV [22,23]. A scoping review conducted in China in 2015 by Biswas et al. revealed notable differences in SB patterns among older adults in care facilities, depending on the level of care provided. According to the review, residents in high-level care facilities, such as nursing homes, spent more time sedentary each day (11.6 h) compared to those in intermediate/mixed-level care facilities (9.5 h). The most common sedentary activity in intermediate/mixed-level care facilities was television viewing, accounting for 26% of daily sedentary time, while napping was the preferred activity in high-level care facilities, accounting for 36% of waking hours [24]. Additionally, more than half of older adults living in LTCFs engaged in sedentary activities for prolonged periods of 30 min or more at a time [25]. The process of institutionalization of older people leads to an increase in SB as instrumental life activities decrease in the residence [26].

As an older person spends more time in an institutionalized setting, their risk of developing SB increases and this leads to a higher degree of muscle mass and strength loss associated with sarcopenia, as well as related health issues [26]. In addition, the quality of life of institutionalized older people decreases with SB and sarcopenia [27]. These findings are particularly concerning, as high levels of SB, in conjunction with chronic health conditions and medication use, have been identified as mediators for the association between obesity and falls among older adults [28].

According to a recent study, 40% of nursing home residents experience feelings of loneliness, and those who feel lonelier and more isolated tend to have lower quality of life. This study has also found that the sense of loneliness among nursing home residents is influenced by their level of mobility [29]. Moreover, a study conducted in 2019 by Meng et al. suggests similar findings: older adults living in LTCFs who experienced loneliness were less likely to participate in activities. This study indicates that interventions that encourage activity engagement among older adults could potentially prevent or reduce frailty and enhance their quality of life [30]. Moreover, reducing SB among older adults in care facilities is critical for maintaining their health and well-being.

The COVID-19 pandemic has further exacerbated the challenges faced by older adults, limiting their daily activities and affecting their overall well-being, especially for those who are institutionalized. As a result of the pandemic, older adults, particularly those who are frail and living in shared housing or LTCFs such as nursing homes, have been subjected to strict confinement measures, such as being confined to their rooms or apartments. These measures, along with social and physical distancing, were implemented as health preventive measures. However, they are likely to have negatively affected the physical and psychological well-being of older adults [31].

In addition, due to quarantine regulations, group activities and other events were restricted in elderly care units, which resulted in increased loneliness and isolation among older people [32]. Loneliness in older adults can have severe negative consequences, including an increased risk of depression, anxiety, cognitive decline, obesity, elevated blood pressure, and even mortality [33,34]. A 2023 study conducted by Escribà-Salvans et al. in Spanish nursing homes reported that in the first year of the pandemic, an increase in falls was observed, resulting in a higher severity of the consequences, i.e., fractures. This could be due to the residents’ activity restriction and consequently their higher SB levels, risk of sarcopenia, osteoporosis and fracture risk [35]. Furthermore, a recent study conducted by Greenwood-Hickman et al. in 2022 compared accelerometer measurements in a group of older adults before and after the COVID-19 pandemic, revealing that the pandemic may have had an adverse effect on older adults’ health [31]. The results indicated that systolic blood pressure was significantly higher in the post-pandemic group, suggesting that the pandemic may have had a negative impact on cardiovascular health. Although accelerometer-measured and self-reported activity levels were not significantly different between the two groups, the post-pandemic group showed trends towards increased sitting time, fewer daily steps, and more self-reported TV time [31]. These findings suggest that the pandemic had the potential to negatively impact the health of older adults, leading to further health problems, especially in those older adults who were already vulnerable due to existing health conditions or social isolation. Thus, it is crucial to prioritize interventions aimed at reducing sedentary time and addressing social isolation to safeguard the well-being of older adults during and after the pandemic [32].

### 1.3. Significance of Reducing SB

The World Health Organization’s (WHO) latest guidelines recommend reducing SB for all age groups, including older adults, individuals with chronic conditions, and those with disabilities [36]. According to a recent systematic review, a reduction of 30 min of sitting time per day can have clinical significance [37]. Interventions aimed at reducing SB can focus on either reducing total daily sedentary time or decreasing the number of prolonged sedentary periods. The Health Measures Survey involving 4935 adults in Canada revealed that breaking up prolonged sedentary time (of 20 min or more) with at least one minute of light, moderate, or vigorous PA was linked to roughly 4% lower triglycerides, 0.6% lower glucose, and 4% lower insulin levels [38]. A study conducted in 2011 by Gardiner et al., examined the feasibility of reducing sedentary time among older adults through intervention strategies such as incorporating breaks from sitting, promoting light-intensity activities, and making environmental changes; the findings suggest that reducing sedentary behaviour is feasible and can lead to positive outcomes in improving older adults’ well-being and reducing the risk of chronic diseases [39]. The results of these studies indicate that even small changes in SB can have a positive impact on metabolic health outcomes.

Understanding how to alter patterns of behaviour is crucial for reducing SB among older adults. A qualitative analysis of focus groups consisting of 31 older adults in assisted living revealed that the main barriers to reducing SB are at the individual level, such as lack of motivation, pain, and fatigue. However, barriers at the organizational or social environment level, including safety concerns, lack of activities outside of business hours, and social norms, were also identified. On the other hand, residents and staff from LTCFs cited physical health, motivation, and safety concerns as significant barriers to sitting less [40]. These findings suggest that designing interventions to reduce SB for older adults who do not live independently should consider unique opportunities to overcome these barriers. Conducting a systematic review in this area will help identify consensus in good interventional practices and contribute to the development of evidence-based guidelines. Ultimately, these efforts will empower older adults to lead healthier, more active lives in LTCFs.

Although several systematic reviews have been conducted on interventions to reduce SB among older people, they only included community-dwelling older adults, while others did not specify the age group or the study setting where the older population was being researched [41,42,43]. To the best of our knowledge, there is no systematic review or meta-analysis on this topic that specifically summarizes the findings of studies on interventions to reduce SB among older people residing in LTCFs. It is important to review this specific group of frail people as the living environment in care facilities can impact SB, as noted in a US-based qualitative study where focus group participants reported increased SB after transitioning from community living to a residential facility. The living environment plays an important role in the SB of older adults; for instance, activities of daily living are usually conducted by the facility staff, making older adults much more sedentary [44]. Hence, the aim of this systematic review will be to collect and compare the evidence on the effects of different interventions aimed at reducing SB among older people living in LTCFs. The research question for this systematic review is as follows: do interventions aimed at reducing SB, compared to usual care, result in a decrease in SB daily time or a reduction in the length of prolonged and uninterrupted sitting bouts in older people living in LTCFs?

## 2. Materials and Methods

### 2.1. Research Methods

We completed the PROSPERO registration for our systematic review protocol to ensure transparency and adherence to best practices. The registration involved providing details about the review, including the research question and objectives. We specified the methodology, search strategy, inclusion/exclusion criteria, and data synthesis plan. We also outlined the anticipated outcomes and the planned approach for data extraction and synthesis. PROSPERO registration enhances transparency, prevents deviations or selective reporting, and promotes collaboration among researchers. The protocol for this systematic review was registered in PROSPERO with the registration number CRD42023404576.

We constructed our research question by using PICO components:Population: older people (60 years old and older) living in LTCFs;Intervention: interventions aimed at reducing SB;Comparison: usual care;Outcome: decrease in SB daily time or reduction in the length of prolonged and uninterrupted sitting bouts.

We employed Boolean operators to enhance the effectiveness and precision of our search strategies. By combining search terms using operators such as “AND”, and “OR”, we were able to refine our searches and retrieve relevant articles. “AND” was used to narrow down the search results by requiring both terms to be present and “OR” was used to broaden the search by including either term. This approach allowed us to systematically identify relevant studies that met our inclusion criteria.

In order to identify relevant MeSH terms for our scientific article, we employed a two-step approach. Firstly, we conducted a comprehensive literature search using the PubMed database. We entered keywords and phrases related to our topic of interest and reviewed the search results. For each retrieved article, we carefully examined the ‘MeSH Terms’ section, which provided standardized medical subject headings assigned to the articles. We noted the MeSH terms that were most closely aligned with our research topic. Secondly, to ensure a thorough exploration of MeSH terms, we also utilized the MeSH database directly. We accessed the MeSH database website and entered our keywords in the search bar. We reviewed the search results and explored the MeSH terms associated with our topic. By examining the details of each MeSH term, including related terms and definitions, we further refined our list of relevant MeSH terms. By employing this dual approach, utilizing both the PubMed database and the MeSH database, we ensured a comprehensive and accurate selection of MeSH terms that accurately represented the scope and focus of our scientific article.

This systematic review will follow the Preferred Reporting Items for Systematic Reviews and Meta-Analyses (PRISMA) guidelines [45].

### 2.2. Types of Studies, Participants and Interventions

In the review, any primary intervention study (including randomized controlled trials, non-randomized controlled trials, and cohort studies), as well as study protocols, will be included. Any non-primary studies (such as reviews) will be excluded.

Only the studies where the participants meet the inclusion criteria (older people, 60+ years of age, living in LTCFs) will be included. Studies that include participants with characteristics stated in the exclusion criteria (older people living in sheltered housing or a residential complex that does not provide daily nursing or social care) will be excluded from the review. 

Any studies that include any interventions with the aim of reducing time spent in SB or shorten the length of prolonged and uninterrupted sitting bouts in older adults living in LTCFs will be searched for. The studies that aim only to increase PA will be excluded, but those studies that target both the reduction in SB and the increase in PA will be included. The interventions will be compared either with no intervention group or with standard care in LTCFs, or with interventions that do not have SB reduction components.

Different comparators or controls might be expected in this systematic review, including usual care, non-intervention control (a group of institutionalized older people who are not given any additional intervention during the time of the study), and a different non-active or PA-based intervention not focused on SB reduction. Studies with before and after designs, with and without follow-up assessments, will be included.

### 2.3. Information Sources

Only peer-reviewed articles will be included in this systematic review. The Cochrane Library, Cochrane Central Register of Controlled Trials (CENTRAL), Ovid MEDLINE, PEDro, ScienceDirect, ClinicalTrials.gov, Google Scholar, and Scopus will be searched. The reference lists of the included articles will also be screened, applying a snowballing approach. If necessary, authors will be contacted to identify any additional unpublished or on-going studies.

### 2.4. Search Strategies

Search strategies using free text terms and controlled vocabulary that describe our target population, intervention, and outcomes will be developed. Medical Subject Headings (MeSH) keywords will be included whenever possible.

The terms for searching will be as follows: “elderly”, “senior(s)”, “older adult(s)”, “elders”, “geriatric(s)”, “geriatric patient(s)”, “old people”, “older-age”, “old age”, “adult(s)”, “older people”, “elderly people”, “60+ adults”, “sedentary behaviour(s)”, “sedentary lifestyle”, “sedentary time”, “physical activity”, “physical activities”, “long term care”, “nursing home”, “care home”, “long-term care facility”, “institution”, and “nursing care”. For this review, date restrictions will not be applied, but there will be language restrictions as only studies in English will be searched. The papers where only the study protocol is provided, as well as the studies with no designated outcomes, will be excluded from the search.

Full search strategies for this systematic review are listed in Table 1 (below).

### 2.5. Selection Process

After searching the databases, abstracts of the studies will be screened by the two primary reviewers according to the protocol’s inclusion/exclusion criteria. Full text articles for the included studies will be retrieved for further screening by two primary reviewers according to eligibility criteria. If necessary, a third reviewer will assess the decisions made at the abstract and full text screening steps and will discuss any discrepancies.

The review will adhere to PRISMA reporting guidelines: the studies excluded at the full text screening step will be listed with the reasons for exclusion, while included studies will undergo quality assessment and data extraction [45].

### 2.6. Data Extraction Process

The data to be extracted will include the following: author name, type and duration of the study, location of the study (country), age and gender of participants, description of the intervention (type of PA, frequency, duration), tools of assessment (scales, devices, or other recording methods used), control group/comparators used, results/measured outcomes, quality (risk of bias), and additional comments.

## 3. Results

The primary outcomes of this review will be SB or time spent sedentary, assessed before and after intervention, using measuring devices (GPS based tracking apps, accelerometers, inclinometers, pedometers, or other devices) and self-reported measures (standardized questionnaires, self-reporting diaries, or other methods). The secondary outcomes will be divided into three categories as follows: mental health (quality of life, depression), physical health (physical function, cardiovascular and metabolic health, adverse events), and social health (social isolation).

The risk of bias of the included studies will be independently assessed by two primary reviewers using the ROBINS-I or Risk of Bias (RoB) 2.0 tool, depending on the type of study [46]. Any disagreements in risk of bias assessment will be resolved through discussion with a third reviewer.

Our data will be grouped according to study design, with separate analysis for randomised designs and non-randomised designs. The characteristics of included studies will be presented in tabular form: the data that are suitable in numbers will be calculated between the intervention and control groups. If it is possible to calculate the treatment effects from the included studies, Review Manager 5 will be used to calculate outcome data. For those outcomes that have the same units of measurements, the pooled mean differences will be calculated, and the standardised mean differences will be calculated for the outcomes with different measurement units. For the data that are suitable in numbers, the key topics will be identified and synthesised narratively. Data related to secondary outcomes will also be assessed if these data are sufficient. If the data are suitable for quantitative analysis, we plan to use the Revman 5 software to conduct a meta-analysis.

A subgroup analysis based on the method used to measure SB will be conducted either objectively using devices or subjectively using self-reported forms. If possible, a subgroup analysis based on the duration of the intervention will be also conducted: the interventions will be classified as those delivered during a single contact with participants or longitudinal interventions.

The strength of evidence of the outcomes will be assessed using the GRADE assessment, which will be based on inconsistency, risk of bias, reporting bias, and imprecision. An overall grade (ranking from very low to high) will show the confidence in the evidence for the outcomes.

## 4. Discussion

Before starting the process of conducting the systematic review, it is important to discuss the main advantages and possible limitations that the research team may face.

### 4.1. Advantages of the Systematic Review

One of the key advantages of this systematic review is its focus on a vulnerable population that is often neglected in research studies. Older people living in LTCFs are at higher risk of developing various health related problems due to their limited mobility and reduced PA levels, as they are more sedentary than those living in community dwelling or assisted living situations [17]. Thus, identifying effective interventions to reduce SB in this population can have significant implications for their health and overall well-being.

Moreover, the inclusion criteria of this review encompass a broad range of designs of primary studies. This approach allows for a more comprehensive evaluation of the available evidence, and the finding can be used to guide the development of evidence-based interventions aimed at reducing SB in this population.

The fact that this systematic review will also adhere to the PRISMA reporting guidelines, which provides a standard framework for reporting systematic reviews and meta-analyses, ensures that the review will be conducted in a transparent and rigorous manner, thereby increasing the reliability and validity of our findings. In addition, the use of the GRADE assessment to evaluate the strength of evidence of the outcomes of this systematic review will provide a clear understanding of the quality and reliability of the available evidence. This approach helps to identify gaps in the literature and highlights areas where further research is needed. It also allows for a more nuanced interpretation of the findings, taking into consideration the quality of the available evidence.

Finally, the findings of this review may have significant implications for clinicians and researchers working in the field of gerontology. By identifying effective interventions to reduce SB in older people, the findings of this review will be able to guide the development of interventions that are tailored to the unique needs and challenges faced by older people living in LTCFs.

In summary, the review’s rigorous methodology, adherence to reporting guidelines, and use of robust tools for assessing risk of bias and strength of evidence will make its findings reliable and trustworthy, thereby contributing significantly to the existing literature on this important public health issue. As far as we are aware, this systematic review will be the first to comprehensively evaluate interventions designed to reduce SB in this specific population group. Although there are existing systematic reviews in this area, their focus is predominantly on older adults living in community settings rather than those residing in LTCFs [27]. Hence, by conducting this systematic review, we aim to fill this research gap and provide insights into effective strategies for reducing SB in older adults living in LTCFs.

### 4.2. Limitations of the Systematic Review

In order to ensure the highest quality of research, it is important to address any potential limitations that may arise in the course of a systematic review. One of the primary concerns is the likelihood of publication bias. This means that studies with positive results are more likely to be published, while those with negative or inconclusive findings are less likely to be published [47]. This can skew our analysis and limit the validity of our findings. Additionally, conducting studies in LTCFs is challenging due to factors such as difficulties in recruitment, communication barriers, and the complex health needs of residents [48]. The difficulty of conducting research in nursing homes is further compounded by the fact that the population being studied is relatively under researched, and this limitation may result in a lower amount of available data to analyse, which could further limit the findings.

To mitigate these limitations, an analysis, such as a funnel plot, will be conducted to investigate potential publication bias. Furthermore, every possible measure will be taken to identify and collect as many relevant articles as possible from various sources to address the limitation of the under-researched population.

## 5. Conclusions

In conclusion, this protocol outlines the systematic review process to be undertaken to investigate the impact of the interventions aiming to reduce SB in older adults residing in LTCFs. The systematic review will involve a rigorous search strategy to identify relevant studies from various databases. We will employ specific inclusion and exclusion criteria to select studies that meet the predefined objectives of the review. Data extraction will be conducted systematically, ensuring that relevant information on study characteristics, participants, interventions, and outcomes is captured. Synthesis of the findings will be performed using appropriate statistical methods and/or narrative analysis. The results will be organized and presented in a clear and structured manner to facilitate the interpretation of the data. Any potential biases and limitations of the included studies will be critically assessed, and the overall quality of evidence will be evaluated using established criteria. The findings of this review will inform healthcare professionals, policymakers, and researchers as well as the general audience about the effects of interventions aiming to reduce SB in this population and highlight the need for targeted interventions.

## Figures and Tables

**Table 1 healthcare-11-01976-t001:** Full search strategies for bibliographic databases.

Bibliographic Database	Search Terms
Cochrane Library	#1 MeSH descriptor: [Sedentary Behaviour] 1 tree(s) exploded #2 (sedentary NEXT (behaviour* or lifestyle* or time*)):ti, ab, kw OR (inactivity):ti, ab, kw OR (screen NEXT (behaviour* or watching* or time* or entertainment*)):ti, ab, kw OR ((sitting or television or TV) NEXT time*):ti, ab, kw (Word variations have been searched) #3 #1 or #2 #4 MeSH descriptor: [Frail Elderly] esxplode all trees #5 MeSH descriptor: [Aged] explode all trees #6 (“elderly” or “senior” or “geriatric” or “adult”): ti, ab, kw OR (older NEXT (adult* or age* or people*)): ti, ab, kw (Word variations have been searched) #7 #4 or #5 or #6 #8 #3 and #7
Ovid MEDLINE	Search via PubMed: ((((“sedentary behaviour”[Title/Abstract] OR “sedentary lifestyle”[Title/Abstract] OR “sedentary time”[Title/Abstract] OR “inactivity”[Title/Abstract] OR “screen behaviour”[Title/Abstract] OR “screen time”[Title/Abstract] OR “screen watching”[Title/Abstract] OR “screen entertainment”[Title/Abstract] OR “sitting time”[Title/Abstract] OR “television time”[Title/Abstract] OR “tv time”[Title/Abstract]) AND “elderly”[Title/Abstract]) OR “senior”[Title/Abstract] OR “geriatric”[Title/Abstract] OR “adult”[Title/Abstract] OR “older adult”[Title/Abstract] OR “older age”[Title/Abstract] OR “older people”[Title/Abstract] OR “aged”[Title/Abstract]) AND “long term care”[Title/Abstract]) OR “nursing home”[Title/Abstract] OR “care home”[Title/Abstract] OR “long term care facility”[Title/Abstract] OR “institution”[Title/Abstract] OR “nursing care”[Title/Abstract]
PEDro	Abstract & Title: Sedentary Subdiscipline: Gerontology
ScienceDirect	Find articles with these terms: “sedentary behaviour”, Title, abstract, keywords: “nursing home”, “elderly”, “senior”, “old adult”, “old people”, “geriatric”, “old age”
ClinicalTrials.gov	Elderly OR Senior OR seniors OR older adult OR older adults OR geriatric OR geriatrics OR old people OR older-age OR old age OR adult OR adults OR older people|Studies With Results | Interventional Studies|Sedentary Behaviour|Older Adult
Google Scholar	(“sedentary behaviour” or “sedentary lifestyle” or “sitting time” or “inactivity”) AND (“old people” or “older people” or “senior” or “elderly” or “older age” or “60+ age” or “60 and over”) AND (“nursing home” or “care home” or “long-term care” or “care facility” or “care institution”)
Scopus	(sedentary or inactivity or “physical activity”) AND (old* or senior or elder* or aged) AND (“nursing home” or “care home” or “long-term care”)

## Data Availability

Not applicable.

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
