# Peer review of "Effectiveness of Interventions for Reducing Sedentary Behaviour in Older Adults Living in Long-Term Care Facilities: A Protocol for a Systematic Review"

_healthcare, 2023, doi:10.3390/healthcare11141976_

Round 1

Reviewer 1 Report

This study is positioned as a protocol research for systematic review. It is believed that this investigation proposes a methodology for systematic examination based on its findings. Notably, the significance of this study is underlined by its focus on older adults residing in Long-Term Care Facilities (LTCFs). Given that older adults in LTCFs are at a higher risk of encountering health-related issues, it is perceived that this population requires more research attention for the prevention and resolution of these problems. It is anticipated that, based on this protocol study, subsequent research could yield academically meaningful results.

The strengths of this study lie in the thorough explanation of its background and objectives, and its well-structured design based on the PRISMA guidelines. Moreover, the study has clearly specified the inclusion and exclusion criteria, and has depicted a highly specific and feasible search strategy. Thus, it is assessed that the study is arranged to allow execution of research as per the protocol. One potential area of improvement could be a more detailed description in the results section, yet considering this study is a protocol research, it seems acceptable to keep the descriptions succinct as they currently are.

In summary, the study holds value as a piece of research that has devised and presented a protocol enabling systematic examination. It's clear that research needs to detail the execution process in such a way that other researchers can carry it out, and this study complies with that requirement. Therefore, I recommend it for publication.

Author Response

The authors thank you for your encouraging comments. The authors have made some minor changes in the results section and they have also added the conclusions section according to the comments of other editors, so now the article is a little bit more detailed as it was suggested by you as well.

Reviewer 2 Report

My comments are in the attached file.

Reviewer 3 Report

Dear Authors,

The topic addressed is interesting and deserves a constructive discussion.

Minor revision

In the protocol as well, I think that ABSTRACT should be in the form of a structured summary. That is, background; objectives; data sources; participants, and interventions; study appraisal and synthesis methods; results; conclusions and implications of key finding.

Author Response

The authors appreciate your comments and have edited the abstract so now it is in the form of a structured summary as it was recommended by you.

Reviewer 4 Report

My heartfelt gratitude for providing me with the opportunity to review the paper I just have have few comments:

Introduction

1.Please more why older adult are risk of SB

2. What are interventions to promote health and welling to avoid SB

Methods.

1. Please add the process how did you add prospero registration

2. Inclusion criteria-Did you add other reviews?

3. How did you get the MESH terms?

4. You have several databases,how did you able to extract artitcles?

Round 2

Reviewer 2 Report

The comments from the first round have been taken into account. Thank you very much. This systematic review protocol could be published.